# Urbanization and Unfavorable Changes in Metabolic Profiles: A Prospective Cohort Study of Indonesian Young Adults

**DOI:** 10.3390/nu14163326

**Published:** 2022-08-14

**Authors:** Farid Kurniawan, Mikhael D. Manurung, Dante S. Harbuwono, Em Yunir, Roula Tsonaka, Tika Pradnjaparamita, Dhanasari Vidiawati, Angelica Anggunadi, Pradana Soewondo, Maria Yazdanbakhsh, Erliyani Sartono, Dicky L. Tahapary

**Affiliations:** 1Division of Endocrinology, Metabolism, and Diabetes, Department of Internal Medicine, Dr. Cipto Mangunkusumo National General Hospital and Faculty of Medicine Universitas Indonesia, Jakarta 10430, Indonesia; 2Metabolic, Cardiovascular, and Aging Research Cluster, The Indonesian Medical Educational and Research Institute, Faculty of Medicine Universitas Indonesia, Jakarta 10430, Indonesia; 3Department of Parasitology, Leiden University Medical Center, 2333 ZA Leiden, The Netherlands; 4Department of Biomedical Data Science, Leiden University Medical Center, 2333 ZC Leiden, The Netherlands; 5Division of Family Medicine, Department of Community Medicine, Faculty of Medicine Universitas Indonesia, Jakarta 10310, Indonesia; 6Makara UI Satellite Clinic, Universitas Indonesia, Depok 16425, Indonesia; 7Center for Sport and Exercise Studies Cluster, The Indonesian Medical Educational and Research Institute, Faculty of Medicine Universitas Indonesia, Jakarta 10430, Indonesia

**Keywords:** urbanization, adiposity, dietary intake, adipokines, young adults, prospective cohort

## Abstract

The substantial increase in the prevalence of non-communicable diseases in Indonesia might be driven by rapid socio-economic development through urbanization. Here, we carried out a longitudinal 1-year follow-up study to evaluate the effect of urbanization, an important determinant of health, on metabolic profiles of young Indonesian adults. University freshmen/women in Jakarta, aged 16–25 years, who either had recently migrated from rural areas or originated from urban settings were studied. Anthropometry, dietary intake, and physical activity, as well as fasting blood glucose and insulin, leptin, and adiponectin were measured at baseline and repeated at one year follow-up. At baseline, 106 urban and 83 rural subjects were recruited, of which 81 urban and 66 rural were followed up. At baseline, rural subjects had better adiposity profiles, whole-body insulin resistance, and adipokine levels compared to their urban counterparts. After 1-year, rural subjects experienced an almost twice higher increase in BMI than urban subjects (estimate (95%CI): 1.23 (0.94; 1.52) and 0.69 (0.43; 0.95) for rural and urban subjects, respectively, P_int_ < 0.01). Fat intake served as the major dietary component, which partially mediates the differences in BMI between urban and rural group at baseline. It also contributed to the changes in BMI over time for both groups, although it does not explain the enhanced gain of BMI in rural subjects. A significantly higher increase of leptin/adiponectin ratio was also seen in rural subjects after 1-year of living in an urban area. In conclusion, urbanization was associated with less favorable changes in adiposity and adipokine profiles in a population of young Indonesian adults.

## 1. Introduction

As a low-middle income country, Indonesia is facing two major health problems. On the one hand, an increasing prevalence of non-communicable diseases such as cardiovascular diseases (CVD), obesity, and type 2 diabetes (T2D) is becoming rampant. While on the other, infectious diseases such as helminth infections, malaria, and tuberculosis are still highly prevalent in some rural areas, resulting in stark differences of these disease patterns between urban and rural settings [1,2].

People residing in urban areas are characterized by relatively high caloric and fat intake compared to their rural counterparts [3]. Moreover, urban people tend to be less physically active [4]. These factors can cause a disruption in energy homeostasis, with a surplus stored in the body as fat [5]. Increasing body fat increases the chance of obesity [6]. Previous studies have shown that higher adipose tissue mass is associated with higher inflammation and insulin resistance [7], which eventually could lead to T2D [8] and CVD [9]. 

Rapid socio-economic development in Indonesia has promoted the migration of people from rural to urban areas to seek a better life [10]. Previous studies have shown that urbanization is associated with new environmental and lifestyle changes that have the potential to put rural individuals at risk of deteriorating metabolic health [11,12,13]. The limitation of these previous studies evaluating the effect of urban-rural environment on metabolic health is their cross-sectional design, which lacks the power to show causality.

The worldwide increase of obesity is not only observed in older populations but also in young adults [14]. Based on the Indonesian National Basic Health Survey 2018, there is a high burden of obesity and prediabetes in the young adult population [15]. As this population constitutes a significant proportion of Indonesians [16], the increase in the prevalence of these diseases may become a major health burden. 

Early problem identification and intervention targeted towards this economically active young adult population in the context of metabolic health could have a great impact on decreasing the incidence rate, or even lowering the prevalence of non-communicable diseases. To this end, we conducted a prospective cohort study to assess the effect of urbanization over time and its contributing factors on the metabolic health profiles of the Indonesian young adult population. 

## 2. Methods

### 2.1. Study Design and Population

This prospective cohort study was conducted on the Depok campus of the University of Indonesia (UI). Freshmen/women UI bachelor students were recruited in this study. Baseline data were collected in the first three months of the start of the academic year, between August-November 2018, while the follow-up sample collection was performed one year later. Subjects’ recruitment was started by providing information about the study during the medical examination of newly arrived students, via social media, and by spreading flyers/leaflets after classes, as well as in student dormitories. A short interview was performed to collect information regarding the areas where the students originated from. Afterwards, a detailed explanation of the study was given to the subjects who agreed to participate and fulfilled the criteria set in this study. After written informed consent, subjects were invited to visit the Makara UI Satellite Clinic to undergo clinical assessment, measurements, and blood sampling. The subjects were classified into the urban group if they were born and lived in urban areas, such as in Jakarta metropolitan areas or in one of the provincial capital cities. The rural group comprised subjects that were originally born and lived in rural areas, defined as the villages that are located at the district levels across Indonesia. Pregnancy and students with previously known diabetes, prediabetes, severe liver or kidney dysfunction, cardiovascular and autoimmune diseases were excluded from the study. This study was approved by the Ethical Committee of Faculty of Medicine Universitas Indonesia (No. 1181/UN2.F1/ETIK/2017). 

### 2.2. Anthropometric Measurements

Body height was measured using a portable stadiometer (SECA Model 213, Seca Gmbh Co., Hamburg, Germany), while body weight and body composition were measured using a Tanita body impedance analyzer (TBF-300A, Tanita Corp, Tokyo, Japan). Body mass index (BMI) was calculated in kg divided by squared height in meters. Three measurements of waist circumference were taken for each subject using an ergonomic circumference measuring tape (SECA Model 201, Seca Gmbh Co., Hamburg, Germany) and according to the WHO standardized protocol. The average of all three measurements was then used for analysis. 

### 2.3. Fasting Blood Glucose, HbA1c, Fasting Insulin, and HOMA-IR Measurement

All clinical measurements and blood samples collection were performed after overnight fasting. Finger prick blood was used for measurement of fasting blood glucose (Accu-Check Performa, Roche Diagnostic GmBH, Germany) and HbA1c (A1c EZ 2.0 HbA1c Analyzer, BioHermes, Wuxi, China) levels. The results of fasting blood glucose (FBG) and HbA1c were used to detect subjects with undiagnosed diabetes and prediabetes that had to be excluded from the study. Serum fasting insulin levels were measured in a certified commercial laboratory (Prodia Lab) by a solid-phase, enzyme-labeled chemiluminescent immunometric assay (Siemens IMMULITE 2000XPi) with an assay range of 2–300 mU/L. For the levels below 2 mU/L, a standardized formula from the instrument manufacturer was used to interpolate the concentrations. Homeostatic model assessment for insulin resistance (HOMA-IR) as a validated measure for whole-body insulin resistance (IR) in humans was calculated using the formula: HOMA-IR = fasting serum insulin × fasting glucose/22.5) [17]. 

### 2.4. Leptin, Adiponectin, and Leptin/Adiponectin Ratio

Serum leptin and adiponectin levels were measured by ELISA using commercial reagents (DuoSet ELISA R&D System) according to the manufacturer’s protocol. Leptin to adiponectin (L/A) ratio, a more sensitive marker for adipose tissue dysfunction, was calculated by L/A = leptin level (ng/mL)/adiponectin level (μg/mL) [18].

### 2.5. Dietary Intake Analysis

One week before the intended measurement date, each subject was informed and instructed on how to make a 3-day food record consisting of two working days and one day during the weekend. For each recording day, all participants were required to write down all of the food and drink they consumed throughout the day. The household servings portion for each meal, food preparation methods, brand name of the foods or beverages if applicable, as well as the addition of sugar, were recorded, as described previously [19]. On the study subjects’ clinical measurement and blood sampling day, a certified dietician performed an interview with the subjects to review the completeness and validity of the food record data. These dietary intake data were then analyzed using NutriSurvey 2007 (EBISpro, Willstatt, Germany) software. The amount of total calorie, carbohydrate, fat, and protein intake for each day were obtained and then averaged for further analysis, as published [20]. 

### 2.6. Physical Activity Analysis

Physical activity was assessed using the adapted Global Physical Activity Questionnaire (GPAQ), which was developed by the World Health Organization [21] and validated for the Indonesian population [22]. This self-reported questionnaire comprised 16 questions that were grouped to collect information regarding physical activity over a typical week in three domains: activity at work, transportation (travel to and from places), and recreational activity [21]. All subjects were asked to fill in the questionnaire based on their one-week activities before the measurement date. According to GPAQ analysis guidelines [23], an estimation of the total weekly volume of moderate and vigorous physical activities (MVPA) was given as Metabolic Equivalent-minutes/week (MET. minutes/week), along with the total time spent on MVPA (minutes/week) and total time of sedentary activities in one week (minutes/week) [24]. Furthermore, based on their total volume and time spent on MVPA, the subject’s physical activity level was classified into three categories (low, moderate, and high) [23].

### 2.7. Statistical Analysis

Continuous variables with normal distribution were presented as mean and standard deviation [mean (SD)]. Meanwhile, non-normally distributed data were presented as geometric mean and 95% confidence interval (geomean (95%CI)) and were log-transformed (log2) for analysis. Linear regression (IBM SPSS Statistics ver. 25) was performed to compare the mean differences of independent variables between two groups at baseline when adjustment for covariates was needed. The chi-square test was used to compare categorical data. Mediation analysis for evaluating the effect of dietary intake components on anthropometry parameter differences between rural and urban group at baseline was performed using PROCESS macro ver. 4.0 for SPSS, as described previously [25]. 

The changes in parameters measured at baseline and 1-year follow-up for each group, and the differences of these changes between urban and rural subjects, were analyzed using linear-mixed model as implemented in the lme4 R package [26]. For each parameter, the covariates used in the linear mixed model were origin (urban/rural), time, and their interaction. The within subject correlation was accounted for using a random-intercepts term. The statistical significance of the effects (i.e., changes from baseline within each group and between groups) were tested using the F-test with Satterthwaite’s degree-of-freedom as implemented in lmerTest [27]. Mediation analysis for the BMI and adipokines changes was performed using 5000 bootstrap samples to obtain the 95% confidence interval for the indirect effect of the covariates. In particular, we evaluated the statistical significance in the decrease/increase of the estimate of the outcome variables after correcting for the changes in certain covariates. Linear mixed model analyses and bootstrapping were performed using R version 4.1.2 in RStudio version 1.4. For all tests, statistical significance was considered at the two-sided 5% level.

## 3. Results

### 3.1. Study Population

A total of 189 (106 urban; 83 rural) subjects were recruited at baseline. For urban subjects, 87.7% originated from Jakarta metropolitan areas, while the rest were from other provincial capital cities. The overall loss to follow-up was 22.1%, leaving 81 urban and 66 rural subjects at the one-year assessment time point. The main reasons for loss to follow-up were refusal to continue (18 subjects/9.4%), could not be contacted (22 subjects/11.6%), and moved to study at another university (2 subjects/1.1%). The proportion of loss to follow-up was similar between rural and urban groups (see flow-chart of the study in Appendix A).

### 3.2. Metabolic Profiles of Urban vs. Rural Subjects at Baseline

Age and proportion of males and females were similar between the rural and urban groups. Adiposity indices (BMI, waist circumference, and fat percentage) were significantly higher in urban compared to rural subjects (mean differences (95%CI) after adjustment for age and sex: 2.81 (1.55; 4.07) kg/m^2^, *p* < 0.001; 6.37 (3.25; 9.50) cm, *p* < 0.001; and 5.07 (2.70; 7.44) %, *p* < 0.001; for BMI, waist circumference and fat percentage, respectively). Moreover, if BMI was grouped based on the WHO cut-off for Asian populations [28], we observed a higher proportion of overweight/obese in urban compared to rural subjects. Conversely, the proportion of underweight subjects was almost three times higher in the rural than in the urban group (Table 1). 

There was no difference in the fasting blood glucose and HbA1c levels between the two groups. Urban subjects had double the HOMA-IR, leptin levels, and L/A ratio than their rural counterparts. The opposite was observed for adiponectin levels. Further adjustment for BMI revealed that the differences remained significant for L/A ratio, while for HOMA-IR, leptin, and adiponectin became not statistically significant (Table 1). 

### 3.3. Dietary Intake and Physical Activity at Baseline

Regarding dietary intake, we observed that urban subjects had significantly higher total calorie, fat, and protein intake compared to their rural counterparts (mean differences (95%CI) after adjustment for age and sex: 162.0 (59.4; 264.7) kcal, *p* = 0.002; 8.2 (3.7; 12.6) gram, *p* < 0.001, and 8.4 (4.7; 12.2) gram, *p* < 0.001), for total calorie, fat, and protein intake, respectively) (Table 1). Additionally, the differences in BMI, waist circumference, and fat percentage between the two groups were slightly attenuated after further adjustment for fat and protein intake, despite remaining statistically significant ((2.22 (0.92; 3.52) kg/m^2^, *p* = 0.001 for BMI; 4.95 (1.74; 8.16) cm, *p* = 0.003 for waist circumference; and 4.38 (1.91; 6.85)%, *p* = 0.001 for fat percentage)). Moreover, mediation analysis showed that fat intake, compared to the other dietary intake components, might be the major driver of the differences in the adiposity profiles between urban and rural subjects at baseline (Appendix A).

Next, we compared the physical activity profiles between the two groups at baseline based on the GPAQ analysis. The results showed that urban subjects had higher total volume and total time spent on MVPA compared to their rural counterparts. However, if these parameters were categorized as low, moderate, or high physical activity levels, no statistically significant differences were observed between the two groups. Meanwhile, for the total time of sedentary activities, we observed lower values for urban compared to rural subjects. (Appendix A).

### 3.4. Effect of Urbanization over Time on Adiposity Profiles, Insulin Resistance, and Adipokines

At follow-up, after one year, both groups experienced an increase in their BMI. When we compared the degree of changes over time, we found that the increase of BMI in rural subjects was almost double what was seen in their urban counterparts (estimate (95%CI) after adjustment for age and sex: 1.23 (0.94; 1.52), *p* < 0.001 and 0.69 (0.43; 0.95), *p* < 0.001, for rural and urban subjects, respectively, P_int_ < 0.01). Although a similar pattern was observed for fat percentage, the difference between the groups did not reach statistical significance (2.18 (1.39; 2.97), *p* < 0.001 in rural subjects vs. 1.33 (0.62; 2.04), *p* < 0.001 in urban subjects, P_int_ = 0.12). Meanwhile, HOMA-IR at one-year follow-up did not change significantly compared to baseline in either rural or urban groups (Figure 1).

Similar analysis was performed for adipokines data, which revealed that both groups had increased leptin levels at 1-year follow-up, with a trend towards a higher increase in rural than urban subjects (Figure 2A, Table 2). Additionally, no changes in the adiponectin levels were observed in urban subjects at the follow-up time point, but a significant decrease was found in the rural subjects (Figure 2B, Table 2). These changes caused no differences in the adiponectin levels between the two groups at 1-year follow-up time point (Appendix A). When L/A ratio was considered, a significant three times higher increase was seen in the rural compared to urban group (Figure 2C, Table 2). After further adjustment with the changes in BMI over time, these changes of leptin, adiponectin, and L/A ratio were attenuated and became non-significant for urban subjects (Table 2).

### 3.5. Effect of Urbanization over Time on Dietary Intake and Physical Activity

The changes over time in two important factors associated with urbanization-related lifestyle, namely, dietary intake and physical activity, were considered next. At the follow-up time point, a significant increase in total calorie, fat, and protein intake was seen in both groups. However, only the increase in protein consumption was significantly higher in rural than in urban subjects ((7.99 (4.42; 11.56), *p* < 0.001 vs. 14.03 (9.95; 18.10), *p* < 0.001, for urban and rural, respectively, P_int_ = 0.03) (see Figure 3). These changes also resulted in the loss of differences in the protein intake between the two groups at the 1-year follow-up time point (Appendix A). Similar to the findings at baseline, adjustment for the increase in fat intake after one year contributed to the largest attenuation of the BMI increase in both groups (in urban: 29.0% vs. 5.8% vs. 1.4%, and in rural: 19.5% vs. 8.9% vs. 7.3%, for fat, protein, and carbohydrate intake, respectively) (Table 3). Although the increase in protein intake was almost twice as high in the rural group compared to the urban group after 1-year, adjustment for protein intake changes did not attenuate the differences in the increase in BMI between the two groups (Appendix A). 

With respect to physical activity, we found a significant decrease in the total volume of MVPA after one year in the urban group only. However, the difference of changes between the two groups was not statistically significant. A similar pattern was also observed for the total time spent on MVPA. Moreover, there was a significantly higher decrease in total sedentary time after one year in the rural group, with a trend for a decrease in the urban group (Appendix A). Furthermore, addition of the physical activity parameters to the model with fat and protein intake did not significantly further attenuate the estimated changes of BMI in either group (Table 3). 

## 4. Discussion

Here, we report the first prospective cohort study in an Indonesian young adult population that evaluated the effect of urbanization on metabolic health profiles. Our study showed that rural subjects had overall better adiposity, insulin resistance, and adipokine profiles compared to their urban counterparts. Importantly, we observed a significantly higher increase in BMI and leptin/adiponectin ratios in the rural subjects migrating to an urban area compared to subjects originating from urban areas.

The higher adiposity indices, proportion of overweight/obese, and whole-body insulin resistance in urban compared to rural residents of Indonesia have been reported before [29]. Unhealthy dietary behavior, such as high intake of calories and fat-dense foods associated with urban living, is thought to contribute to the higher adiposity profiles [3]. Indeed, we confirmed this pattern of dietary intake in our study. Although all further adjustments with total calorie, fat, or protein intake attenuated the anthropometric differences between rural and urban groups, our study showed that fat intake contributed the most. Additionally, the longitudinal follow-up to see how urban lifestyle affects metabolic health in those migrating from rural areas compared to urban residents, first confirmed a significant increase of BMI after one-year follow-up in both groups, as seen in previous studies, showing that the majority of freshmen gain weight during their first year of university life [30,31]. The increase of total calorie, fat, and protein intake after one year in both groups might partially explain these changes in BMI. Our study also implicated fat intake changes as the dietary intake component that might be the major contributor for BMI increase after one year in both groups. Significantly, the rural group experienced almost a twice higher increase in BMI than the urban group. Although a significantly higher protein intake was observed in rural compared to urban group at one year follow-up, this could not explain the differences in the BMI increase between the two groups. Interestingly, previous studies have shown that higher meat or meat-products intake, which mostly consist of protein and fat, is associated with more weight gain independent of the total calorie intake [32].

Another factor contributing to adiposity profiles is the level of physical activity [33], as this promotes burning of calories, leading to negative energy balance and subsequently less probability for fat deposition [34]. In our study, at baseline, we found that rural group had lower total volume and time spent on MVPA with a higher sedentary time compared to urban group. This suggests that physical activity does not explain the differences observed in BMI. However, it has to be noted that studies of physical activity in rural and urban areas can be influenced by factors such as the level of education, ethnicity, and tools utilized [35,36]. In our study, the questionnaires used at baseline, which took place when the study subjects had already arrived in urban area, might not truly reflect the subjects’ level of physical activity during their residence in rural areas. At the 1-year follow-up time point, we found no significant differences in the changes of total volume and time spent on MVPA between the two groups. As for fat or protein intake, physical activity did not explain the higher gain in BMI seen in the rural group. 

The addition of the physical activity parameters into the model with the adjustment of dietary intake also could not explain the higher increase of BMI in rural compared to urban group. This result implies that with similar changes of dietary intake and physical activity within one year, rural subjects experienced bigger changes in BMI than their urban counterparts. Hence, other factors, such as the gut microbiome [37] or epigenetic changes [38], could potentially influence the adiposity changes in the rural population upon migration to urban areas. Other factors that could potentially influence the changes of weight or BMI in our study subjects, as shown in previous studies, are psychological stress [39] and socioeconomic-cultural backgrounds [40,41]. These factors were not evaluated in our study.

The increase of BMI in both groups, if continued for the long term, could potentially cause obesity and induce other metabolic and cardiovascular diseases. In other cases, if the BMI increase does not lead to obesity, the distribution of body fat caused by the weight gain also needs to be considered. Previous studies have shown that Asian populations tend to have higher cardiometabolic risks compared to Caucasian populations with the same levels of BMI, in particular, due to central obesity or visceral adiposity [42,43]. These risks would potentially be higher in the rural group as a substantial number of subjects are underweight. Several studies showed that individuals with previous malnourished condition have an increased risk of obesity later in life, especially if they adopt unhealthy lifestyles [44,45]. Moreover, individuals that have experienced a double burden of malnutrition or undernutrition in early life followed by later overweight/obesity, also pose a substantially enhanced risk of non-communicable diseases (NCDs) [46].

The observed differences in insulin resistance and adipokine levels at baseline between urban and rural groups aligned with the findings from previous studies, showing a higher HOMA-IR and lower adiponectin levels in urban compared to rural population [47,48]. Moreover, after adjustment for BMI, the differences in HOMA-IR, leptin, and adiponectin were no longer statistically significant, indicating a major contribution by BMI. Interestingly, after 1-year living in an urban area, a significant decrease in adiponectin levels, along with a significant increase in L/A ratio, was observed in rural groups compared to the urban group. These changes were attenuated after adjustment for the changes in BMI. This finding shows that rural subjects also experienced worse changes in the adipokine profiles, which was partially mediated by the changes in BMI. It also indicates that there might be other factors than BMI, potentially contributing to the changes in adipokines in the rural subjects after 1-year living in an urban area. As shown from previous studies, gut microbiota has been associated with changes in leptin and adiponectin levels in response to a high-fat diet [49,50].

In our study, although urban subjects had a significant increase in BMI levels after one year, this did not cause changes in adiponectin levels compared to the significant decrease observed in the rural group. These changes even led to the loss of the differences in adiponectin levels between the two groups at 1-year follow-up time point. These findings showed that rural individuals tend to be more vulnerable in their metabolic parameters upon changes in BMI. 

Leptin and adiponectin have opposite effects on subclinical inflammation and insulin resistance. Leptin upregulates pro-inflammatory cytokines such as tumor necrosis factor-α and interleukin-6, while adiponectin has anti-inflammatory properties [18]. Adipose tissue dysfunction, marked by higher leptin and lower adiponectin levels, has been associated with insulin resistance and the incidence of T2D [51]. However, in our study, there were no significant changes in insulin resistance in the groups studied after one year of residence in an urban area. The relatively short follow-up period and preserved pancreatic beta-cell function in the young adult population might potentially explain this [52]. 

The longitudinal study design, inclusion of several metabolic health parameters, and incorporation of dietary intake and physical activity measurements were several strong points of our study. There is only one previous prospective cohort study known to the author that has evaluated the effect of urbanization on CVD risk factors and major NCDs [12]. However, this study did not incorporate dietary intake analysis and measurement of biological metabolic markers, such as insulin resistance index, leptin, and adiponectin. Our study also observed the importance of fat intake contribution in the increase of BMI in both the freshmen urban group and the rural individuals who recently migrated to an urban area. Previous study evaluating this freshmen weight gain only took into account eating behavior changes but did not perform detailed dietary intake analysis [53]. 

However, the relatively small number of subjects and short duration of follow-up could be considered as limitations in our study. The addition of tools to evaluate the quality of dietary intake, such as the Healthy Eating Index and the utilization of health technology devices like an accelerometer to assess physical activity more objectively, could provide more accurate data in future studies. Additionally, the inclusion of psychological stress assessment and questionnaires or tools to accommodate the evaluation of the socio-economic and cultural aspects would also result in a more comprehensive data for future research. Moreover, investigation of the gut microbiome, epigenetic changes, as well as immunological factors, might shed more light on the mechanisms that underlie rapid changes in the metabolic profiles upon urbanization. 

In conclusion, the findings in our study show that adoption of an urban lifestyle could potentially cause poorer metabolic health changes in rural individuals who migrate to an urban area. Our findings, in part, complement a previous study that showed the rising BMI in residents of increasingly urbanizing rural areas in low-middle income countries is due to an increase in low-quality diet [54]. However, it also indicates that there is a more rapid increase in BMI of subjects arriving from rural areas that could not be explained by either diet or physical activity. Therefore, further studies are needed, as it is important for policymakers to design innovative approaches to prevent this negative effect of urbanization in young adult population, with particular attention to those migrating from rural areas.

## Figures and Tables

**Figure 1 nutrients-14-03326-f001:**
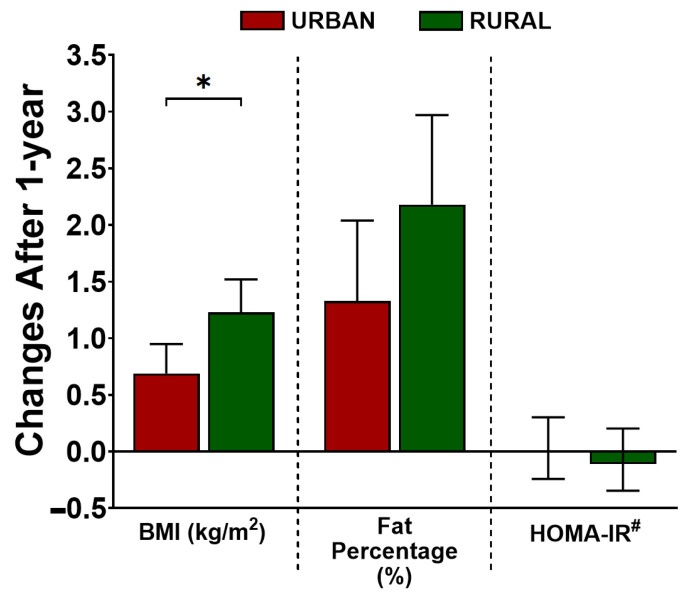
Changes of BMI, fat percentage, and whole-body insulin resistance (HOMA-IR) in urban and rural subjects after 1-year of living in an urban environment. The changes are presented as estimate and 95% confidence interval (95%CI). The changes in each group and the differences of changes between the urban and rural group for each parameter were analyzed using a linear-mixed model, adjusted for age and sex. The *p*-value depicted in the figure represents the *p*-value for interaction (P_int_), the level of significance in the differences of changes between the two groups. * *p* < 0.05. ^#^ HOMA-IR was log-transformed (base 2) for analysis. The estimates (95%CI) were back-transformed (2^β^) and presented as a multiplicative scale compared to baseline. BMI: body mass index; HOMA-IR: homeostatic model assessment for insulin resistance.

**Figure 2 nutrients-14-03326-f002:**
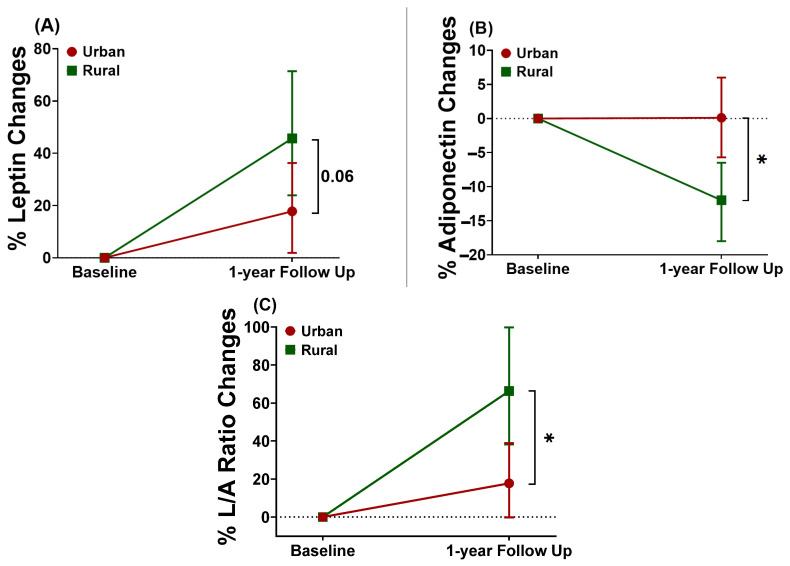
Changes of leptin levels (**A**), adiponectin levels (**B**), and leptin-adiponectin (L/A) ratio (**C**) in urban and rural subjects after 1-year of living in an urban environment. The changes are presented as estimate and 95% confidence interval (95%CI). The changes in each group and the differences of changes between urban and rural group for each parameter were analyzed using a linear-mixed model, adjusted for age and sex. All parameters were log-transformed (base 2) for analysis. The estimates (95%CIs) were back-transformed (2^β^) and presented as percent changes compared to baseline. The *p*-value depicted in the figure represents the *p*-value for interaction (P_int_), the level of significance in the differences of changes between the two groups. * *p* < 0.05.

**Figure 3 nutrients-14-03326-f003:**
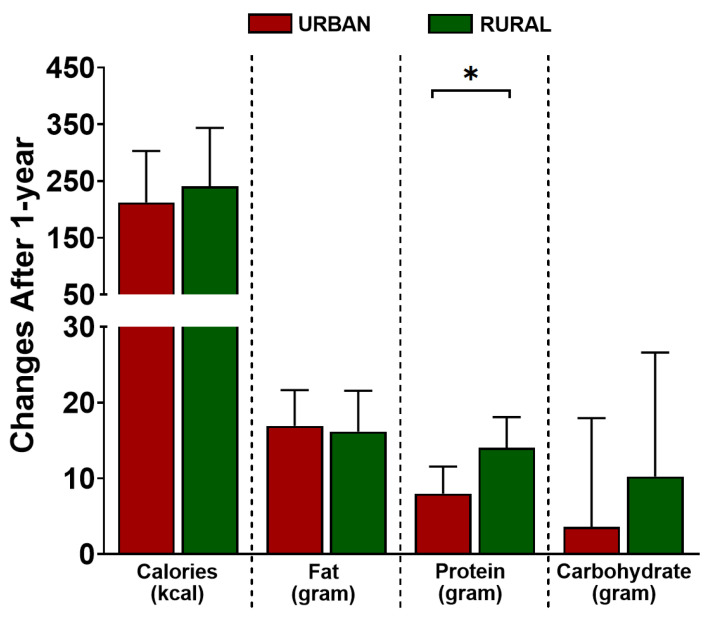
Changes of calorie-, fat-, protein-, and carbohydrate intake in urban and rural subjects after 1-year of living in an urban environment. The changes are presented as estimate and 95% confidence interval (95%CI). The changes in each group and the differences of changes between urban and rural group for each parameter were analyzed using a linear-mixed model, adjusted for age and sex. The *p*-value depicted in the figure represents the *p*-value for interaction (P_int_), the level of significance in the differences of changes between the two groups. * *p* < 0.05.

**Table 1 nutrients-14-03326-t001:** Baseline characteristics of the study population.

Variables	Urban N = 106	Rural N = 83	*p* Values ^#^ (Adjusted for Age and Sex)	*p* Values ^#^ (Adjusted for Age, Sex, and BMI)
Age, yrs old (mean, SD)	18.4 (0.7)	18.6 (0.7)	0.09	
Sex, n male (%)	39 (36.8)	31 (37.3)	0.94	
BMI, kg/m^2^ (mean, SD)	22.9 (5.0)	20.0 (3.2)	**<0.001**	
BMI grouping, n (%) - Underweight (<18.5) - Normoweight (18.5–22.9) - Overweight (23–24.9) - Obese (≥25.0)	14 (13.2) 50 (47.2) 17 (16.0) 25 (23.6)	28 (33.7) 41 (49.4) 7 (8.4) 7 (8.4)	**0.001**	
Waist circumference, cm (mean, SD)	78.5 (12.8)	72.1 (8.2)	**<0.001**	
Fat percentage, % (mean, SD)	28.2 (9.1)	22.8 (8.3)	**<0.001**	
FBG, mg/dL (mean, SD)	87.1 (8.2)	86.7 (7.8)	0.54	
HbA1c, % (mean, SD)	5.1 (0.4)	5.1 (0.3)	0.24	
Fasting insulin ^†^, IU/mL	5.3 (4.3–6.6)	2.9 (2.2–3.8)	**0.001**	0.06
HOMA-IR ^†^	1.1 (0.9–1.4)	0.6 (0.5–0.8)	**0.001**	0.06
Leptin ^†^, ng/mL	11.6 (9.7–13.8)	6.9 (5.3–9.1)	**<0.001**	0.07
Adiponectin ^†^, µg/mL	4.1 (3.7–4.5)	4.9 (4.4–5.3)	**0.02**	0.19
Leptin-Adiponectin (L/A) Ratio ^†^	2.9 (2.3–3.5)	1.4 (1.1–1.9)	**<0.001**	0.03
Dietary intake, mean (SD) - Total calories, kcal - Fat, gram - Protein, gram	1444 (335) 52 (15) 50 (14)	1289 (422) 44 (16) 41 (13)	**0.002** **<0.001** **<0.001**	**0.009** **0.01** **0.001**
- Carbohydrate, gram	193 (55)	179 (73)	0.08	0.06

^†^ Not normally distributed continuous variables, presented as geomean (95%CI) and log transformed for analysis. ^#^ Analyzed with linear regression for continuous variables and Chi-square test for categorical variables. The *p*-values shown in bold represent the statistically significant differences with *p*<0.05. BMI: body mass index; FBG: fasting blood glucose; HOMA-IR: homeostatic model assessment for insulin resistance.

**Table 2 nutrients-14-03326-t002:** Mediation analysis of the effect of changes in BMI overtime on the leptin, adiponectin, and L/A ratio in urban and rural subjects at 1-year follow-up.

Variables ^†^	Adjusted for Age and Sex		P_int_		Adjusted for Age, Sex, and BMI			P_int_
	Urban	Rural		Urban	Indirect Effect ^#^	Rural	Indirect Effect ^#^	
Leptin	0.24	0.54	0.06	0.09	−0.25; −0.07	0.33	−0.29; −0.12	0.12
	(0.03; 0.45)	(0.31; 0.78)		(−0.11; 0.29)		(0.10; 0.55)		
	*p* = 0.03	*p* < 0.001		*p* = 0.38		*p* = 0.005		
Adiponectin	0.002	−0.19	0.003	0.04	0.01; 0.06	−0.12	0.03; 0.10	0.008
	(−0.08; 0.09)	(−0.29; −0.10)		(−0.04; 0.12)		(−0.22; −0.03)		
	*p* = 0.97	*p* < 0.001		*p* = 0.34		*p* = 0.008		
L/A ratio	0.23	0.73	0.006	0.06	−0.30, −0.08	0.45	−0.39; −0.17	0.02
	(−0.003; 0.47)	(0.47; 1.00)		(−0.16; 0.28)		(0.20; 0.70)		
	*p* = 0.05	*p* < 0.001		*p* = 0.60		*p* < 0.001		

^†^ All variables were analyzed using a linear mixed model on log transformed data, presented as estimate and 95% confidence interval. ^#^ Indirect effect of BMI on the variables analyzed, obtained by performing bootstrapping with 5000 iterations and presented as its 95% confidence interval. BMI: body mass index; L/A ratio: leptin/adiponectin ratio; P_int_: *p*-value for interaction.

**Table 3 nutrients-14-03326-t003:** Mediation analysis of the effect of changes in dietary intake and physical activity over time on the changes of BMI at 1-year follow-up in both urban and rural subjects.

Model ^†^	Urban				Rural				P_int_
	Estimate (95%CI)	*p* Values	% Changes ^††^	Indirect Effect ^#^ (95%CI)	Estimate (95%CI)	*p* Values	% Changes ^††^	Indirect Effect ^#^ (95%CI)	
**Adjusted for age and sex**	0.69 (0.43, 0.95)	<0.001			1.23 (0.94; 1.52)	<0.001			0.007
**Model with changes in dietary intake**									
(+) Total calories intake	0.55 (0.28; 0.28)	<0.001	−20.3	−0.13 (−0.33; −0.02)	1.02 (0.71; 1.32)	<0.001	−17.1	−0.15 (−0.52; −0.04)	0.02
(+) Carbohydrate intake	0.68 (0.42; 0.93)	<0.001	−1.4	−0.01 (−0.12; 0.03)	1.14 (0.85; 1.43)	<0.001	−7.3	−0.09 (−0.34; 0.01)	0.02
(+) Fat intake	0.49 (0.20; 0.78)	<0.001	−29.0	−0.20 (−0.47; −0.04)	0.99 (0.67; 1.31)	<0.001	−19.5	−0.24 (−0.54; −0.06)	0.01
(+) Protein intake	0.65 (0.38; 0.93)	<0.001	−5.8	−0.04 (−0.22; 0.08)	1.12 (0.78; 1.45)	<0.001	−8.9	−0.11 (−0.43; 0.14)	0.02
(+) Fat and protein intake	0.50 (0.21; 0.79)	<0.001	−27.5	−0.19 (−0.48; 0.02)	1.04 (0.70; 1.37)	<0.001	−15.4	−0.19 (−0.53; 0.05)	0.007
**Model with changes in physical activity**									
(+) Total volume of MVPA	0.68 (0.42; 0.95)	<0.001	−1.4	−0.01 (−0.14; 0.05)	1.23 (0.94; 1.52)	<0.001	0.0	0.0 (−0.10; 0.04)	0.007
(+) Total minutes of MVPA	0.68 (0.42; 0.95)	<0.001	−1.4	−0.01 (−0.15; 0.06)	1.23 (0.94; 1.52)	<0.001	0.0	0.0 (−0.10; 0.05)	0.007
(+) Total sedentary time	0.70 (0.44; 0.96)	<0.001	1.4	0.01 (−0.03; 0.10)	1.26 (0.94; 1.58)	<0.001	2.4	0.03 (−0.11; 0.22)	0.007
**Model with changes in dietary intake and physical activity**									
(+) Fat and protein intake and total volume of MVPA	0.48 (0.19; 0.78)	0.001	−30.4	−0.21 (−0.50; −0.01)	1.10 (0.75; 1.44)	<0.001	−10.6	−0.13 (−0.49; 0.15)	0.003
(+) Fat and protein intake and total minutes of MVPA	0.49 (0.19; 0.78)	0.001	−29.0	−0.20 (−0.51; −0.01)	1.09 (0.75; 1.44)	<0.001	−11.4	−0.14 (−0.52; 0.14)	0.003
(+) Fat and protein intake and total sedentary time	0.51 (0.22; 0.80)	<0.001	−26.1	−0.18 (−0.47; 0.001)	1.15 (0.79; 1.50)	<0.001	−6.5	−0.08 (−0.47; 0.21)	0.002

^†^ All variables as an additional adjustment for age and sex, and all analyses were performed using linear-mixed model. The group of covariates used for model adjustment were shown in bold. ^††^ Proportion of changes in the estimate of the model compared to the model adjusted for age and sex only. ^#^ Indirect effect of covariate(s) on BMI, obtained by performing bootstrapping with 5000 iterations and presented as its 95% confidence interval. BMI: body mass index; MVPA: moderate-vigorous physical activity; P_int_: *p*-value for interaction.

## Data Availability

The data that support the findings of this study are available from the corresponding author (F.K.) upon reasonable request.

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
