# Peer review of "Urbanization and Unfavorable Changes in Metabolic Profiles: A Prospective Cohort Study of Indonesian Young Adults"

_nutrients, 2022, doi:10.3390/nu14163326_

Round 1
Reviewer 1 Report
There is an interesting paper related to the effect of changes in nutrition and physical activity on BMI and related biochemical parameters. However, significant improvement is needed before this publication can be accepted for publication.
Editorial work is still needed and “wording “ should be improves, e.g. „fell short of statistical significance” should not be used in scientific paper.
All data after 1-year follow-up must be presented (on similar way as baseline data)
Tables 2,3, S1 – why it is called “Pathway analysis” ? No data related to “pathways” are presented
Fig.2. There in no reason to present changes using logarithmic scale. It does not improve the interpretation of the data
Most study participants are women. No data on their hormonal status, pregnancy etc are presented.
More precise description of participants recruitment is needed.
The tables, Figures and data description (under Results) should be improved. The biological significance of observed changes as well as of statistical analyses should be discussed.
Different factor which could affect changes in BMI of subjects from rural area must be discussed.
The novelty of the study should be underlined
Reviewer 2 Report
Authors have written an interesting article about important topic. However, there are few things to be considered. Like authors also point out; measuring the quality of the diet and also actual physical activity (with better questionnaires and possible also an accelometer) woud make it even better.
Major concerns:
Table 2 is hard to read when lines change in the middle of numbers. Please improve.
Introduction and discussion
For example lines 341-343: Animal products and animal derived protein has been associated to weight gain/obesity and many other diseases. See for example:
-eating meat-free diet is likely to lead consuming less calories https://pubmed.ncbi.nlm.nih.gov/21616194/
-plant-based eating may lead to higher resting metabolic rate https://pubmed.ncbi.nlm.nih.gov/8177051/
-meat consumption is associated to weight gain even independent of calories https://pubmed.ncbi.nlm.nih.gov/20592131/
-with children eating animal products is associated to being overweight https://pubmed.ncbi.nlm.nih.gov/20237136/
Please discuss if this could explain weight also in this study.
Minor concerns:
Lines 32-33: It seems that this sentence is missing a verb?
Line 73: There is extra space after "University of"
Lines 175-178: Should results of adiposity indices be given in the same order as they are first mentioned (BMI, waist circumference and fat percentage)? Please correct.
Line 324: Extra space after word "BMI"
Line 396: Extra space between two sentences.
Reviewer 3 Report
In my opinion, the manuscript is well organized and presented.
The introduction section is not long but contains all the preliminary information necessary to create a solid background for better comprehension of the study.
The methods are correct and well described.
Results are well presented and discussed, with a proper number of tables and figures.
The discussion section provides the authors' explanations for the observed results.
In my opinion, this manuscript is suitable for publication in its present form. Only a general check for the English language is required, in particular for the use of appropriate scientific words.
Round 2
Reviewer 1 Report
Line 22 - “a longitudinal study” – it should be clearly indicated that You performed a 1-year follow-up study
Line 78 – it “as many subjects” – did you want to say as many as possible ??? . It must be improved and proper way of description of subject requiretment should be used
Because of the physiological differences in anthropometric parameters between men and women, as well waist circumference, fat content (in general for the same BMI fat content is different in men and women) and in leptin levels (higher in women than men) data for men and women should be presented separately. If the observed changes in anthropometric and related biochemical parameters are similar in men and women, for specific statistical analyses the data could be grouped.
Fig.2 . From biological point of view, it is important to see observed changes in the concentrations of leptin and adiponectin not their logarithmic transformation. Logarithmic transformations can be used in different statistical analyses (according to the rules) but statistical analyses can only help to recognized specific biological effects.
Table 2. leptin and Adiponectin are independent variables but their ratio is not, therefore should not be used for such type of statistical analyses (the results are not informative from biological point of view).
In addition, the idee of presented statistical analysis should be better explained. It is not clear why the effect of BMI should be different in urban and rural groups, as well as how the calculated “indirect effect” of changes in BMI on leptin and adiponectin concentrations should be interpreted.
